# Intracranial Solitary Fibrous Tumor: A “New” Challenge for PET Radiopharmaceuticals

**DOI:** 10.3390/jcm11164746

**Published:** 2022-08-14

**Authors:** Angela Sardaro, Paolo Mammucci, Antonio Rosario Pisani, Dino Rubini, Anna Giulia Nappi, Lilia Bardoscia, Giuseppe Rubini

**Affiliations:** 1Section of Radiology and Radiation Oncology, Interdisciplinary Department of Medicine, University of Bari “Aldo Moro”, 70124 Bari, Italy; 2Section of Nuclear Medicine, Interdisciplinary Department of Medicine, University of Bari Aldo Moro, Piazza Giulio Cesare 11, 70124 Bari, Italy; 3Radiation Oncology Unit, S. Luca Hospital, Healthcare Company Tuscany Nord Ovest, 55100 Lucca, Italy

**Keywords:** solitary fibrous tumor, hemangiopericytoma, PET/CT, fluorodeoxyglucose, fluorocholine, non-FDG radiopharmaceuticals

## Abstract

Solitary fibrous tumor (SFT) of the central nervous system, previously named and classified with the term hemangiopericytoma (HPC), is rare and accounts for less than 1% of all intracranial tumors. Despite its benign nature, it has a malignant behavior due to the high rate of recurrence and distant metastasis, occurring in up to 50% of cases. Surgical resection of the tumor is the treatment of choice. Radiotherapy represents the gold standard in the case of post-surgery residual disease, relapse, and distant metastases. In this context, imaging plays a crucial role in identifying the personalized therapeutic decision for each patient. Although the referring imaging approach in SFT is morphologic, an emerging role of positron emission tomography (PET) has been reported in the literature. However, there is still a debate on which radiotracers have the best accuracy for studying these uncommon tumors because of the histological or biological heterogeneity of SFT.

## 1. Introduction

In 2021, the latest and most recent version of the 5th edition of WHO Classification of Tumors of the Central Nervous System (CNS5) was published. One of the most important innovations is the introduction of a new classification of tumor types and subtypes, thanks to novel diagnostic technologies such as DNA methylome profiling. Focusing on mesenchymal, non-meningothelial tumors, the hybrid term solitary fibrous tumor/hemangiopericytoma (SFT/HPC), previously used in the 2016 WHO classification, has been abandoned in favor of the use of the term solitary fibrous tumor (SFT) alone to fully comply with the nomenclature of soft-tissue pathology [1].

SFTs are rare mesenchymal malignancies of soft-tissue cells with a high percentage of recurrence. In particular, intracranial SFTs constitute 2.5% of meningeal neoplasms and less than 1% of all intracranial tumors [2]. At present, magnetic resonance imaging has a well-established role in the identification and diagnosis of this rare tumor. However, because of SFTs’ high tendency to have locoregional recurrences and distant metastases, in recent years, multimodality imaging, particularly positron emission tomography/computed tomography (PET/CT), has shown an increasingly significant role, thanks to the possibility of obtaining both metabolic and morphological information in a single scan. Furthermore, it is a non-invasive and total-body examination, with high sensitivity and specificity [3,4].

We aimed to present a brief critical overview of the role of PET/CT in diagnosing, local disease relapse, or distant metastases evaluation in the treatment response to surgery and/or radiotherapy and in the follow-up of patients affected by intracranial SFT.

## 2. Solitary Fibrous Tumor

SFT is a very rare and uncommon neoplasm, accounting for only 1.6% of all central nervous system (CNS) tumors. In the 2016 World Health Organization (WHO) classification of CNS tumors, SFTs and HPCs constituted a single disease entity, known as SFT/HPC. However, in the last recent update of WHO CNS5, the term “hemangiopericytoma” was deleted to emphasize the biological similarities within tumor types and to align with the soft-tissue pathology nomenclature [1].

This tumor was firstly known by the term hemangiopericytoma, coined by Stout and Murray in 1942, to highlight the origin from the capillary and postcapillary venules of pericytes. It can occur in any anatomical body region with the presence of capillaries and a typical localization involves the meninges of the dura mater, with an incidence of 16–33% of SFTs in the head or neck [5]. From a histological point of view, these tumors are characterized by spindle mesenchymal cells with a rich vascular component. Usually, the presence of hypercellularity, hemorrhage and/or necrosis, pleomorphic nuclei, and foci of dedifferentiation suggests a malignant behavior [6]. The tumor immunophenotype can be characterized by CD34 and smooth-muscle-actin positivity and by S100 protein, CK pool, and desmin negativity [7]. In accordance with previous immunohistochemical features, the new WHO CNS5 reported that STFs show NAB2 and STAT6 gene fusion as well as STAT6 overexpression [2].

Even many years after the end of the first (line) treatment, locoregional relapses, as well as distant metastases, are very common, and a prompt diagnosis could be crucial for better patient management. However, probably due to SFTs’ rarity, there are no clear recommendations on the best imaging method to evaluate them [8]. The literature is characterized by small-sample-size studies and case series, showing that complete surgical resection with clear margins is the treatment of choice. However, one year after the stand-alone surgery option, a very high percentage of disease relapse, rating from 88 to 100%, may occur as a result of tumor infiltration in adjacent vascular structures with a worsening of survival [9]. Stereotaxic radiosurgery has emerged as a promising adjuvant treatment to reduce the relapse rate of the disease [2]. In an updated overview on radiation therapy options for intracranial HPC, Ciliberti and colleagues showed that postoperative radiotherapy (RT) can lead to a consistent decrease in local recurrences, moving from 88% to 12.5%, and longer relapse-free survival [9]. In addition, when the lesion is close to critical anatomical structures, stereotactic radiotherapy can be considered. However, even after a wide surgical resection or a combined surgery–radiotherapy treatment, recurrences and metastases can subsequently occur in up to 50% of cases [10,11].

In this scenario, imaging can play a crucial role in potentially guiding the best therapeutic decision and in establishing tailored treatment.

Although the referring imaging approach in SFTs is morphologic, it may fail in differentiating between scar tissue and viable tumor in post-treatment evaluation. For this purpose, PET has demonstrated a promising role [12,13].

## 3. Search Strategy

A literature search was conducted on the Medline (PubMed) database including all articles published up to 30 June 2022. The following keywords were entered to rule the search: “intracranial solitary fibrous tumor” AND “positron emission tomography”, AND “PET” AND “nuclear medicine”, “hemangiopericytoma” AND “positron emission tomography” AND “PET” AND “nuclear medicine”, “intracranial hemangiopericytoma” AND “positron emission tomography” AND “PET” AND “nuclear medicine”. Only articles edited in English in the last 10 years were included in this review. After reading the abstracts, some articles were excluded because they did not meet the goal of our review in evaluating the use of PET/CT in patients affected by intracranial SFT/HPC. For the same reason, some articles were not considered in the final draft after reading the full text. To identify supplementary eligible articles, additional references were searched from the retrieved review articles. Due to the rarity of SFT, most of the articles in the literature are represented by case reports and interesting images. The main characteristics of the included studies are detailed in Table 1.

## 4. Evidence-Based Medicine of PET/CT in Intracranial Solitary Fibrous Tumor/Hemangiopericytoma

In the last decade, PET/CT emerged as a non-invasive, whole-body diagnostic tool with an important role in diagnosis, staging, detecting possible disease relapses and distant metastases, differentiating between scar and viable tissues, and in the follow-up of patients affected by SFT. To the best of our knowledge, this is the first literature review that focuses on the role of PET/CT in intracranial SFTs. This topic could reveal to be great interest in identifying the best accurate PET radiopharmaceutical to diagnose and follow up this rare tumor. Indeed, despite its benign nature, SFT has a malignant behavior, with a characteristic high rate of locoregional relapse and of distant metastases due to its hypervascularity.

To date, most of the authors referred to ^18^F-FDG as the main PET radiotracer used in this oncological field to assess glucose metabolism activity, usually increased in neoplastic cells. ^18^F-FDG is intracellularly trapped after GLUT-transporter uptake and hexokinase phosphorylation in glucose-6-phosphate. The hypercellularity of SFT lesions may support its application in these cases. However, the intense physiological uptake of the brain makes adequate ^18^F-FDG PET imaging of intracranial tumors a challenge [21]. In some cases of SFT/HPC, ^18^F-FDG PET may be helpful to differentiate necrotic from viable tumors, with both hypoenhancing on CT but with different metabolic activities on PET, significantly influencing the treatment decision [28]. Generally, FDG uptake in SFT/HPC cells is decreased compared with the surrounding tissue, while intense ^18^F-FDG hypermetabolism may suggest the presence of a more malignant variety of SFT, with a significant impact on prognosis [22]. The relatively low SUVmax values in ^18^F-FDG PET studies could help to differentiate SFT from other malignancies. In addition, a differential diagnosis with respect to other tumors may be facilitated by a dual-tracer technique to compare different image patterns developed by two PET radiotracers. For example, low glucose metabolism in contrast to high ^11^C-Methionine uptake may help to differentiate HPC from meningioma [29].

The high probability of developing metastases linked to the malignant behavior of this mesenchymal tumor makes ^18^F-FDG PET/CT a potential ideal tool to ensure adequate whole-body examination and to correctly guide therapeutic decision making with a tailored approach [8].

In Wu et al.’s case report, ^18^F-FDG PET/CT was performed to restage a 25-year-old man affected by a malignant SFT of the right occipital lobe primarily treated with multiple craniotomies and postoperative conformal RT. PET/CT showed both intracranial disease recurrence and, above all, massive bilateral pulmonary lesions and multiple bone metastases [14].

Cheung et al. confirmed the fundamental role of ^18^F-FDG PET/CT in identifying distant metastases from SFT/HPC describing a case of a 67-year-old woman treated with the resection of a paravertebral soft-tissue mass with histological diagnosis of HPC. After eight years, MRI detected three histologically proven SFT/HPC lesions of the right posterior occipital calvarium, while ^18^F-FDG PET/CT showed multiple hypermetabolic lesions involving lymph nodes and bone from the calvarium to the sacrum [15].

Furthermore, the literature reports a case of 41-year-old man surgically treated for a cerebellar HPC. After twenty-two years, the patient experienced multiple renal and pulmonary metastases, pathologically proven to be HPC, and a local intracranial recurrence, all of them surgically treated. Two years later, abdominal CT revealed a pancreatic tumor, confirmed with whole-body ^18^F-FDG PET/CT with intense radiopharmaceutical uptake. This lesion was surgically removed and histopathological examination confirmed the diagnosis of HPC, with similar pathological findings to those of the original cerebellum HPC [16].

However, in some cases, ^18^F-FDG PET/CT may not prove to be adequately reliable in accurately detecting all SFT/HPC lesions, since low–moderate glucose metabolism in both primary and metastatic lesions can be observed. Recently, Liu Xiao et al. reported a case of a 40-year-old patient affected by intracranial STF/HPC with a left renal metastatic lesion showing only mild ^18^F-FDG uptake and homogeneous contrast enhancement after contrast-medium administration [17].

Similarly, Hayenga and his colleagues reported a case of a 34-year-old woman affected by HPC of the right cerebellopontine angle (CPA) who underwent surgical treatment combined with postoperative RT. After 3 years of disease-free interval, ^18^F-FDG PET/CT was performed showing low FDG avidity both in intracranial and extracranial recurrences of the spinal canal [18]. Consistent with these reports, Yasen and colleagues reported negative FDG uptake in the presence of HPC recurrence and metastasis. Namely, a woman with bilateral-frontal-cerebral-convex and parafalx HPC was surgically treated and underwent ^18^F-FDG PET/CT six years later, demonstrating high-density lesions on CT images without FDG uptake and low-density solid lesions without FDG uptake in the frontal lobe parafalx and in the right posterior and left outer lobes of the liver, respectively. The histological examination of calvarium and hepatic lesions showed homogeneous features of HPC tumor [19]. These are examples of SFT/HPC lesions with low glucose metabolism, almost similar to background FDG activity, suggesting that ^18^F-FDG PET/CT should not be the only diagnostic tool to be used in this set of patients, especially for the detection of craniospinal-axis SFT/HPC [18].

Another issue that deserves consideration and that may limit ^18^F-FDG PET/CT performance is the FDG heterogenous uptake between primitive SFT/HPC and metastatic lesions. In this regard, Grunig and colleagues reported a case of a 46-year-old patient with high FDG uptake in primary-dura HPC and liver metastases, but moderate radiopharmaceutical uptake in a leg-muscle tissue lesion [20].

In relation to these ^18^F-FDG PET/CT limitations, current literature debates which is the most accurate PET radiopharmaceutical to adequately study this rare benign, but with malignant behavior, tumor with possible histological or biological heterogeneity.

Some authors highlighted that SFT/HPC shows different spectral patterns on MR spectroscopy, with dominant Choline expression [30,31]. This feature suggested the potential usefulness of radiolabeled Choline as an ideal PET radiopharmaceutical in SFT/HPC detection [21,32,33]. Choline is a precursor of the biosynthesis of cell-membrane phospholipids, which is increased in the most malignant tumors, including SFTs. In fact, once intracellular uptake is performed by specific transporters, Choline is phosphorylated by choline-kinase in phosphorylcholine and incorporated into the cell membrane [34,35,36]. Considering the anatomical proximity with brain parenchyma, a further advantage of ^18^F-FCH PET/CT is the very low brain uptake with a favorable tumor-to-background ratio with a better image contrast. The low physiological distribution of Choline in normal white and grey matter may help to distinguish different brain tumors with increased uptake from benign lesions and radiation necrosis with the lowest uptake [36]. In the case report by Sardaro et al., ^18^F-Fluorocholine (^18^F-FCH) PET/CT showed superior performance compared with ^18^F-FDG PET/CT in a 69-year-old patient with two local recurrences of malignant orbital SFT, playing a crucial role in differentiating between small recurrences/pathological lesions and scar tissue in brain parenchyma (Figure 1) [7]. The favorable Choline tumor-to-background ratio [37] allows better and well-defined visualization of the lesion with intracranial localization to be obtained [38].

Similarly, Jehanno et al. reported a case of a 50-year-old male with a previous history of surgically treated intracranial HPC who experienced a local recurrence eighteen months later. After partial resection, the authors performed both preoperative ^18^F-FDG PET/CT and ^18^F-FCH PET/CT for the treatment planning of residual disease located along the optic nerve. ^18^F-FDG PET/CT did not show any significant uptake. Conversely, ^18^F-FCH PET/CT precisely detected an intense radiopharmaceutical uptake of residual disease in the retro-orbital region, more properly guiding the following RT [21].

Some authors reported SFT lesions immunohistochemically and histologically characterized by a high expression of somatostatin receptor (SSRs), especially subtype 2, well studied with somatostatin-receptor molecular imaging methods. Lavacchi and colleagues reported a case of an intracranial SFT/HPC with distant metastasis studied with ^111^In Pentetreotide, a gamma-emitter radiolabeled somatostatin analogue, able to target the highly expressed SSRs in anecdotal SFT lesions, with potential theragnostic implication. In Lavacchi et al.’s case, high radiopharmaceutical uptake was observed both in primitive and distant metastases. Namely, a 64-year-old female affected by cranial-posterior-fossa SFT was restaged 18 years after surgical treatment due to an ^111^In Pentetreotide scan having detected multiple intracranial recurrences and hepatic, renal, and pulmonary metastases with avid tracer uptake [22]. Similarly, high SSR expression in intracranial and extracranial lesions was detected with a ^111^In-Pentreotide scan in a 54-year-old woman with intracranial HPC of the right optic nerve sheath and bone distant metastases. In the same patient, low FDG uptake in the aforementioned sites confirmed the unreliability of ^18^F-FDG PET/CT in some cases [23]. Furthermore, ^68^Ga -Dodecane tetra-acetic acid tyrosine-3-octreotate (^68^Ga-DOTATATE) is a PET radiopharmaceutical usually used for neuroendocrine-tumor imaging and now increasingly considered as an adjunctive imaging tracer for multiple solid neoplasms, including mesenchymal tumors [39]. Similarly to ^111^In-Pentetreotide, ^68^Ga-DOTATATE is a positron-emitter radiolabeled somatostatin analogue able to reveal a high concentration of surface SSRs, with the advantages of superior spatial resolution and accuracy, faster acquisition and lower radiation exposure [24]. In particular, Hung et al. found negligible FDG uptake but intense ^68^Ga-DOTATATE avidity of pulmonary metastases in a 68-year-old woman affected by primitive intracranial HPC [24].

In addition to FDG and choline, the literature reports the use of other PET radiotracers for the identification of this uncommon neoplasm. In some cases, HPC/SFTs may show high ^68^Ga-Prostate-Specific Membrane Antigen (^68^Ga-PSMA) avidity [40]. PSMA is a type II integral transmembrane glycoprotein, known as folate-hydrolase 1 or glutamate-carboxypeptidase II, with neuropeptidase activity. It was found to be surprisingly overexpressed in the neovasculature endothelium of some brain tumors, such as gliomas with significant angiogenic activity, whereas low PSMA expression can be found in tumor cells or healthy brain. Low PSMA uptake in the normal brain parenchyma and its high tumor-to-background ratio allows an accurate localization of intracranial lesions with PSMA overexpression to be performed, which seems not to be related to the type of brain malignancy. The literature reports higher PSMA-uptake in high-grade gliomas and metastatic brain tumors than in central-nervous-system lymphoma and radiation necrosis, while there are no data about the differential diagnosis with SFT/HPC because of the rarity of this tumor [25,36,41]. Patro and colleagues described intense ^68^Ga-PSMA avidity in all hepatic and bones metastases in a 53-year-old woman with primitive right-posterior-cranial-fossa HPC, in contrast with the low glucose metabolism of these lesions, probably due to PSMA overexpression in tumor neovasculature [25].

Interestingly, Zhang et al. showed an intense ^68^Ga-Fibroblast-activation-protein inhibitor (^68^Ga-FAPI) (compared with low ^18^F-FDG uptake) in a 23-year-old female patient with SFT/HPC of the right frontal lobe. Fibroblast activation protein (FAP) is strongly overexpressed in the stroma of human cancers, including SFTs, and its quinoline-based FAP inhibitor can be internalized after binding to the FAP enzymatic domain. This suggests the potential role of this new PET radiotracer with a specific target for FAP, which can be overexpressed in this mesenchymal tumor [26,42]. The Acetate property of being a fatty acid precursor and thus an indirect biomarker of fatty acid synthesis can be exploited. In fact, ACT is converted to Acetyl Co-A, which is incorporated into cholesterol and fatty acids. Jong et al. reported a single case of dual-tracer ^11^C-acetate (^11^C-ACT) and ^18^F-FDG PET /CT in a 47-year-old male patient with intracranial HPC, showing significantly higher ^11^C-ACT uptake of bone metastases than FDG uptake, probably due to the over-expression of fatty acid synthase in this kind of tumors [27,43].

## 5. Conclusions

PET/CT could play a fundamental role in diagnosis, staging, post-treatment evaluation (surgery, radiotherapy, chemotherapy), disease relapses and distant metastases detection, and follow-up of patients affected by intracranial SFT, a benign but inherently aggressive tumor.

However, the literature still debates which type of PET radiopharmaceutical could guarantee the best accuracy for correctly and promptly guiding the management of this set of patients.

^18^F-FDG PET/CT shows an extremely heterogeneous behavior, with modest or low radiopharmaceutical uptake in most cases of single- or dual-tracer studies, variable sensitivity and low reliability.

Compared with ^18^F-FDG PET/CT, ^18^F-FCH PET/CT appears to be superior in detecting intracranial SFTs thanks to the histological and biological features of SFT and a favorable tumor-to-background ratio.

Other radiopharmaceuticals labeled with Gallium 68 also demonstrate a promising role in SFT-relapse or distant-metastases detection.

The current literature shows that ^18^F-FDG is the most widely used radiopharmaceutical in this set of patients. However, although fully aware that numerous studies are needed to identify which radiotracer to use based on the biological and histological characteristics of this rare tumor, our limited experience unquestionably reveals the superiority of ^18^F-FCH PET/CT, compared with ^18^F- FDG PET/CT, for the study of patients with intracranial SFT. ^18^F-FCH PET/CT can play a significant role in providing information for both local recurrence and distant metastases.

Therefore, for all the aforementioned reasons, it would be desirable to include PET/CT as a diagnostic, sensitive, non-invasive method in the guidelines for the management of SFT.

## Figures and Tables

**Figure 1 jcm-11-04746-f001:**
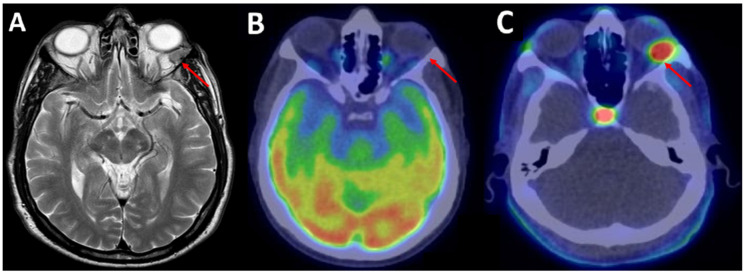
A 69-year-old male patient with a second loco-regional recurrence of left supraorbital solitary fibrous tumor. (**A**) A brain MRI scan revealed on axial MRI T2c+ a rounded lesion on the lateral side of the left orbit, strongly suspected of disease relapse (red arrow). (**B**) Two weeks later, ^18^F-FDG PET/CT showed no radiopharmaceutical uptake in the left supraorbital region (red arrow). (**C**) Conversely, after seven days, ^18^F-FCH PET/CT showed intense uptake in the aforementioned lesion (red arrow; SUVmax 6.8).

**Table 1 jcm-11-04746-t001:** Main characteristics of the included studies of intracranial SFTs/HPCs with probable distant metastases studied using PET/CT or scintigraphy with ^111^In-Pentreotide (*n* = 16).

Case	Authors	Year	Age, Sex	Intracranial Primitive Site	Metastatic Sites	Radiopharmaceuticals	Qualitative and Semiquantitative Uptake Level
1	Z. Wu et al. [14]	2015	25, M	Right occipital lobe	Lungs, bones	^18^F-FDG	Mild–moderate SUVmean 4.9—SUVmax 8.1
2	H. Cheung et al. [15]	2018	67, F	Right posterior occipital calvary	Paravertebral, bones, lymph nodes	^18^F-FDG	Mild *
3	K.P. Cheng et al. [8]	2017	41, F	Intracranial meninges	Bones	^18^F-FDG	Intense *
4	T. Hiraide et al. [16]	2012	41, M	Cerebellum	Kidneys, lungs, pancreas	^18^F-FDG	Intense *
5	X. Liu et al. [17]	2021	40, M	Fronto-parietal	Kidney	^18^F-FDG	Mild, SUVmax 3.17
6	H. N. Hayenga et al. [18]	2019	34, F	Right cerebellopontine angle	Thoracic spine	^18^F-FDG	Low *
7	A. Yasen et al. [19]	2020	62, F	Frontal cerebral convex, parafalx	Liver	^18^F-FDG	Absent
8	H. Grunig et al. [20]	2021	46, F	Intracranial dura	Liver, muscles	^18^F-FDG	High–moderate SUVmax 9.0
9	Sardaro et al. [7]	2021	69, M	Left orbit	/	^18^F-FDG	Absent
						^18^F-FCH	Intense, SUVmax 6.8
10	Jehanno et al. [21]	2019	50, M	Right spheno-orbital region	/	^18^F-FDG	Low, SUVmax 3.5
						^18^F-FCH	Intense, SUVmax 5.9
11	Lavacchi et al. [22]	2020	64, F	Posterior fossa	Liver, kidneys, lungs	^111^In-Pentreotide	Intense *
			35, M	Falx cerebri	Liver	^18^F-FDG	Intense *
12	G. Kota et al. [23]	2013	54, F	Right optic nerve sheath	Bones	^18^F-FDG	Low *
						^111^In-Pentreotide	Intense *
13	T. Hung et al. [24]	2016	68, F	Not specified	Lungs	^18^F-FDG	Minimal *
						^68^GA-DOTATATE	Intense *
14	K.C. Patro et al. [25]	2018	53, F	Right posterior cranial fossa	Bones, liver	^18^F-FDG	Low *
						^68^Ga-PSMA	Intense *
15	Zhang et al. [26]	2021	23, F	Right frontal lobe	/	^18^F-FDG	Low, SUVmax 1.6
						^68^GA-FAPI	Intense, SUVmax 30.9
16	I. Jong et al. [27]	2013	47, M	Not specified	Bones	^18^F-FDG	Mild *
						^11^C-Acetate	Intense *

Abbreviations: FDG = fluorodeoxyglucose; SUVmean = average standardized uptake value; SUVmax = maximum standardized uptake value; FCH = fluorocholine; PSMA = prostate-specific membrane antigen; DOTATATE = dodecane tetra-acetic acid tyrosine-3-octreotate; FAPI = fibroblast-activation-protein inhibitor. Annotation: * = semiquantitative parameters not known.

## Data Availability

Not applicable.

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
