# Peer review of "Intracranial Solitary Fibrous Tumor: A “New” Challenge for PET Radiopharmaceuticals"

_jcm, 2022, doi:10.3390/jcm11164746_

Round 1

Reviewer 1 Report

Dear Authors,

congratulations for this interesting and well written paper about a very uncommon disease charachterized by a challenging behaviour.

The role of methabolic imaging in this setting is well investigated and the review is substantially well conducted.

English language is linear and the bibliography is complete.

My only concern is about the conclusions. PRobably the authors, based also on their experience, should try to give a more robust suggestion on what would be the most used tracer in clinical practice.

Author Response

Thank you very much.

Reviewer 2 Report

This is the review of an Intracranial solitary fibrous tumor visualized by several PET tracers and the author aimed at indicating the best PET tracer for the diagnosis of SFT.   

The author should add the mechanism of accumulation for each PET tracer in more detail.

In table 1, the author showed the uptake level of PET tracers to SFT. Please define the level of uptake and it is good to add the reported SUV for each case in the sentence.

 Please refer to what will be the major differential diagnosis of SFT in each tracer.

Author Response

Thank you very much.
